# Association between the Beighton Score and Stress Ultrasonographic Findings of the Anterior Talofibular Ligament in Healthy Young Women: A Cross-Sectional Study

**DOI:** 10.3390/jcm11071759

**Published:** 2022-03-22

**Authors:** Takuji Yokoe, Takuya Tajima, Nami Yamaguchi, Yudai Morita, Etsuo Chosa

**Affiliations:** Division of Orthopaedic Surgery, Department of Medicine of Sensory and Motor Organs, Faculty of Medicine, University of Miyazaki, 5200 Kihara, Kiyotake, Miyazaki 889-1692, Japan; kingt2@hotmail.com (T.T.); nami_yamaguchi@med.miyazaki-u.ac.jp (N.Y.); yudai_morita@med.miyazaki-u.ac.jp (Y.M.); chosa@med.miyazaki-u.ac.jp (E.C.)

**Keywords:** ankle lateral ligament, joint instability, ultrasonography, women

## Abstract

The Beighton score (BS) is widely used to evaluate generalized joint laxity. However, the association between the BS and lateral ankle laxity is unclear. This study compared the ultrasonographic (US) findings of the anterior talofibular ligament (ATFL) between high- (≥6) and low- (≤3) BS groups of healthy young women. The ATFL lengths of healthy young women were measured in the stress and nonstress positions using the previously reported technique from March 2021 to January 2022. The ATFL ratio (ratio of stress to nonstress ATFL length) was used as an indicator of lateral ankle laxity. The anterior drawer test (ADT) was performed. The correlation between the BS and US findings was also examined. A total of 20 (high-BS group) and 61 (low-BS group) subjects with a mean age of 23.8 ± 1.0 years were included. The high-BS group showed a higher grade of ADT than the low-BS group. No significant differences were found in the nonstress and stress ATFL lengths and ATFL ratio (1.10 ± 0.05 vs. 1.09 ± 0.05, *p* = 0.19) between the groups. No correlation was found between the BS and US findings. In conclusion, this study did not detect significant differences in the US findings of the ATFL between the high- and low-BS groups.

## 1. Introduction

Generalized joint laxity (GJL) is a condition in which most synovial joints have a range of motion beyond the normal limitations [1,2]. Although several scoring systems for evaluating GJL, such as the Hospital Del Mar criteria and Rotes-Querol scoring system, have been reported [3], the Beighton score (BS) is the most commonly used and validated method for the assessment of GJL [2,4,5]. Many previous studies have assessed the association between GJL and postoperative outcomes and the risk of recurrent instability, especially in knee and shoulder joints [6,7]. With respect to the influence of GJL on ankle joints, some authors have demonstrated that GJL is an independent risk factor for recurrence and poor outcomes after the modified Broström procedure for chronic lateral ankle instability (CLAI) [8,9]. However, no studies have clarified the causes of unfavorable clinical outcomes in patients with GJL. Fundamentally, the BS may not accurately reflect the ankle laxity due to its scoring system, which does not include the shoulder, hip or ankle joints [10]. In the shoulder joint, Whitehead et al. reported a poor correlation between the BS and specific measures of shoulder joint laxity [11]. In addition, compared with other joints, there is a lack of studies investigating the effect of GJL on the incidence of ankle injuries, including ankle sprains and CLAI [12,13].

Stress ultrasonography (US) of the anterior talofibular ligament (ATFL) has been demonstrated to be an effective and reliable procedure for the diagnosis of CLAI [14,15,16]. Song et al. evaluated the effect of GJL on the US findings of ATFL, reporting that the ATFL lengths on US in both stress and nonstress positions were significantly longer in the high-BS group (≥5) than in the low-BS group (<5) (*p* < 0.001) [12]. In contrast, Yokoe et al. reported that the influence of GJL on lateral ankle laxity may differ by sex based on their evaluation of the ATFL ratio (ratio of stress to nonstress ATFL length) on stress US [13]. They also found that the normal value of the ATFL ratio was significantly greater in healthy ankles of the female subjects than in male subjects (1.09 ± 0.04 vs. ± 1.07 ± 0.04, *p* = 0.001). Given that young women tend to have greater laxity of the ankle joint and a higher BS than young men [2,13], the influence of GJL on lateral ankle laxity may not be significant in young women. Therefore, a higher cut-off value of the BS may be rational when comparing US findings of the ATFL between high- and low-BS groups of young women in order to clarify the influence of GJL on native lateral ankle laxity. In addition, CLAI was reported to occur nearly twice as often in female athletes as in male athletes (32% vs. 17%) [17]. Thus, it would be beneficial to understand the association between the BS and US findings of the ATFL in healthy young women, as clinicians may use the healthy ankle as a reference when assessing patients with CLAI by stress US.

The purpose of this study was to evaluate the association between BS and stress US findings of the ATFL in healthy young female subjects. The hypothesis of the working group was that BS would not correctly reflect the lateral ankle laxity in this population.

## 2. Materials and Methods

### 2.1. Study Design and Study Population

This cross-sectional study was approved by an institutional review board (Approval NO. O-0669). Healthy female volunteers ≥20 years old were consecutively recruited into a single institute from March 2021 to January 2022. All participants gave informed written consent before participating in the study. Ankles were excluded as follows: a history of ankle sprain, episodes of giving way of the ankle, a history of surgery of the foot or ankle, foot or ankle pain at the time of recruitment, ankle deformities (flatfoot, cavus foot and hindfoot malalignment), inflammatory arthritis, such as rheumatoid arthritis, and Ehlers-Danlos or Marfan syndrome. In addition, ankles were also excluded when the absence of ATFL, lax and wavy ATFL, or avulsion fracture of the distal fibula was detected by US [14,18].

The presence of GJL is historically considered by a BS of ≥4 or ≥5 [2,19]. However, no evidence-based cut-off value of the BS exists to detect GJL [20]. Therefore, to clarify the relationship between the BS and lateral ankle laxity, patients with a score of 4 or 5 were excluded. The high-BS group was defined as subjects with a BS of ≥6, while the low-BS group was defined as subjects with a BS of ≤3. Of the 98 subjects, a total of 81 were finally included after excluding those with a BS of 4 or 5 (*n* = 13), a history of bilateral ankle sprains (*n* = 3), or a history of bilateral ankle surgeries (*n* = 1). A total of 20 subjects (20 ankles) and 61 subjects (61 ankles) were included in the high- and low-BS groups, respectively (Figure 1).

The demographic data included the age, gender, height, weight, body mass index (BMI), side of the ankle (right or left) and foot size. The foot size was defined as the length from the longest toe to the tip of the heel that was measured with a tape measure in the standing position. The assessment of the BS and manual anterior drawer test (ADT) were performed by a senior orthopedic surgeon prior to the US examination. US pictures of the ankle were obtained in the nonstress position (resting position) and the stress position (manual maximal internal rotation), as reported by Yokoe et al. [13]. When bilateral ankles did not meet the exclusion criteria, the dominant ankle was evaluated and included in the study. The dominant ankle was defined as the one used to kick a ball. If a unilateral ankle met the exclusion criteria, the contralateral ankle was evaluated. US evaluations were performed by a certified orthopedic surgeon who was blinded to the subjects’ BS.

### 2.2. The BS

The presence or absence of GJL was assessed based on the BS [4]. The scoring system comprises five objective measurements of joint mobility, four of which are measured bilaterally. A subject receives one point if the little finger hyperextends over 90°, if the thumb touches the volar aspect of the forearm, if the elbow hyperextends over 10°, if the knee hyperextends over 10°, and if the palm completely touches the floor with forward flexion of the trunk while keeping the knees straight. The total score ranges from 0 to 9 points, with a higher score indicating increased laxity. A goniometer was used to measure the extension angle of the elbow and knee joints.

### 2.3. Manual ADT

Manual ADT of the ankle was performed with the subject in the supine position. The knee joint was flexed, and the ankle joint was sustained in 10–15° plantarflexion. The subject was instructed to relax before the performance of ADT. While grasping the heel of the examined ankle with one hand and stabilizing the distal tibia with the other hand, the ankle was anteriorly drawn until no further movement was recognized. The results were classified into three grades: Grade 1, a stable joint; Grade 2, partially unstable; Grade 3, completely unstable [15]. The intra-rater reliability of the ADT was confirmed prior to the initiation of this study by calculating Cohen’s kappa coefficient (κ). The κ value was 0.83.

### 2.4. Stress US Evaluation of the ATFL

US pictures were obtained with an ALOKA ARIETTA 850 US apparatus (HITACHI, Tokyo, Japan) using a linear probe (L64 probe, 18–5 MHz). The stress US procedure was reported previously [13]. US pictures were taken in two positions: the resting position (nonstress ATFL) and the manual maximal internal rotation position (stress ATFL) (Figure 2). Nonstress ATFL images were taken first. The subject was in a sitting position with one leg hanging from the edge of the examination table (resting position). The transducer was placed over the ATFL and was parallel to the sole of the foot. The subject was then instructed to relax the ankle muscles with the ankle joint in 10–20° plantarflexion. The ATFL length was measured as a linear distance from the origin to the insertion of the ATFL. The origin and insertion points of the ATFL were identified as bony landmarks to ensure standardization of the ATFL in a manner previously reported [21]. A static shot was obtained when confirming the bony landmarks. Thereafter, a stress ATFL image was obtained. The subject was first instructed to position in the aforementioned resting position (naturally plantarflexion), and the examiner manually applied maximal internal rotation with varus talar tilt to the ankle (by grasping the heel of the subject). The internal rotation with varus talar tilt in plantarflexion is a better procedure for evaluating lateral ankle laxity [22,23].

The ATFL length was measured as a linear distance from the origin to the insertion of the ATFL, in the same manner as that for nonstress ATFL images. The anterolateral aspect of the lateral malleolus was identified as the ATFL origin, and the peak of the talus was used as the insertion point. The peak of the talus also represents the anterior aspect of the lateral talar articular cartilage and the lateral neck of the talus. These bony landmarks can be identified as hyperechogenic points [24] and were confirmed to ensure that the talar insertion was consistently selected at a reference point across images. Based on the obtained findings, the ATFL ratio was calculated. A static shot was obtained when confirming the end point of the stretched ATFL.

The intra-rater reliability of the US findings was confirmed prior to the present study by calculating the intraclass correlation coefficients (ICCs). The ICC (1,2) for the nonstress ATFL length, stress ATFL length and ATFL ratio were 0.91 (95% confidence interval (CI): 0.90–0.92), 0.88 (95% CI: 0.86–0.89) and 0.93 (95% CI: 0.92–0.94), respectively. The standard error of measurements for the nonstress ATFL length, stress ATFL length and ATFL ratio were 0.51, 0.80 and 0.01, respectively.

### 2.5. Statistical Analysis

Data analysis was performed with the SAS software (JMP Pro, ver. 15.2.0; SAS Institute, Cary, NC, USA). A *p* value < 0.05 was considered statistically significant. The results were reported as mean values with 95% confidence intervals (CIs). The Shapiro–Wilk test was used to confirm whether or not the data were normally distributed. Student’s *t*-test was conducted to compare the weight, foot size and US variables, and the Mann–Whitney U test was performed for the age, height, BMI and BS, according to the Shapiro–Wilk test. The chi-square test was used to compare the results of the side of the ankle and ADT. Spearman’s rank correlation coefficient was used to assess the correlation between the BS and US findings. The strength of the correlation of the rank coefficients was defined as follows: strong, 0.70–1.0; moderate, 0.40–0.69; weak, 0.20–0.39 [25].

Sample size calculation was performed with G* power (version 3.1.9.7). According to the results reported by Song et al. [12], there was a significant difference in the ATFL length between the patients with and without GJL (*p* < 0.001). In the present study, a high-BS group and low-BS group were included at a 1:3 ratio. According to the study of Cohen et al. [26], with an effect size of 0.8, a minimum of 17 and 51 subjects were needed in the high- and low-BS groups to provide 80% power at two-tailed α of 0.05.

## 3. Results

The mean age of the low- and high-BS groups was 23.7 ± 2.1 (range, 20–33) years and 24.3 ± 1.8 (range, 21–27) years, respectively (*p* = 0.14). The mean BS in the low- and high-BS groups was 1.6 ± 1.1 (95% CI, 1.3–1.9; range, 0–3) and 6.8 ± 1.0 (95% CI, 6.3–7.3; range, 6–9), respectively (*p* < 0.001). There were no significant differences in the patient characteristics between the groups except for in the BS (Table 1).

### 3.1. The Comparison of the ADT and US Findings between the Low- and High-BS Groups

The results of the comparison of the ADT and US findings between the high- and low-BS groups are shown in Table 2. The high-BS group showed a significantly higher rate of Grade 2 ADT than the low-BS group (55.0 vs. 27.9%, *p* = 0.03). There were no significant differences in the US findings between the groups as follows (high-BS vs. low-BS group): the nonstress ATFL length, 18.1 ± 1.1 (95% CI, 17.6–18.6) vs. 18.2 ± 1.5 (95% CI, 17.8–18.6) (*p* = 0.99); the stress ATFL length, 20.0 ± 1.4 (95% CI, 19.3–20.6) vs. 19.9 ± 1.7 (95% CI, 19.4–20.3) (*p* = 0.58); the ATFL ratio, 1.10 ± 0.05 (95% CI, 1.08–1.13) vs. 1.09 ± 0.05 (95% CI, 1.08–1.10) (*p* = 0.19).

### 3.2. The Correlation between the BS and US Findings

A correlation between the BS and nonstress ATFL length was not detected (r = 0.01, *p* = 0.90) (Figure 3), nor was any correlation detected between the BS and stress ATFL length (r = 0.07, *p* = 0.52) (Figure 4) or between the BS and ATFL ratio (r = 0.15, *p* = 0.19) (Figure 5).

## 4. Discussion

The most important finding of the present study was that no significant differences in the stress US findings of the ATFL were identified between the low- and high-BS groups of healthy young female subjects. In addition, there was no correlation between the BS and ATFL ratio in this population.

Pacey et al. reported that GJL was not associated with an increased risk of ankle injuries during sports activities [27]. Sueyoshi et al. found that young female athletes with recurrent ankle sprains had a higher BS than those with a history of single ankle sprain [28]. The interaction between GJL and LAS or CLAI has remained controversial. Several authors have reported that GJL is a predictor of worse clinical outcomes following the modified Broström procedure for CLAI [8,9,10]. These authors suggested augmentation to the Broström procedure or reconstruction for patients with GJL. According to a study reporting the consensus of the ESSKA-AFAS Ankle Instability Group [29], 60% of experts preferred reconstruction to repair in patients with GJL for the surgical treatment of CLAI. The presence of GJL is an important factor when orthopedic surgeons make preoperative planning or counsel patients who undergo surgery for the treatment of CLAI. At present, the BS is the most widely used tool to evaluate the presence of GJL in clinical research and practice. Based on the results of the present study, however, the BS has no correlation with lateral ankle laxity in a young female population. Song et al. assessed the association between the GJL and US findings of the ATFL in uninjured ankle joints [12]. The study found no significant difference in the length change between resting and stress ATFLs on US in comparison with patients without and with GJL (BS ≥ 5) (*p* = 0.08), which was comparable to our results. However, the authors reported that both nonstress and stress ATFL lengths were significantly longer in patients with GJL than those without GJL (*p* < 0.001), which were not relevant to our findings. In the present study, the effect sizes of the nonstress and stress ATFL lengths were 0.08 and 0.06, respectively, suggesting that at least there were no significant differences in the ATFL lengths between the high- and low-BS groups, as Song et al. reported. These different results of US evaluations may be mainly due to the different techniques used for stress in US evaluations. In the present study, the stress ATFL length was measured in plantarflexion while applying maximal manual supination with internal rotation. The ATFL mostly elongates in plantarflexion with supination [23,30], and supination with internal rotation was reported to better detect lateral ankle ligaments’ insufficiency [22]. Based on the results obtained by Song et al., the mean ATFL ratio of the patients with and without GJL was 1.04 and 1.06, respectively. These values are markedly smaller than those obtained in our study, indicating that elongation of the ATFL was not sufficient in the stress position. Differences in the age groups, the definition of the high-BS group and patient characteristics may also have influenced the differences in the US findings of the ATFL.

In the present study, there was statistically no correlation between the BS and US findings of the ATFL. However, these findings should be interpreted with caution because the power for the correlation between the BS and nonstress ATFL length, stress ATFL length and ATFL ratio were 0.05, 0.10 and 0.27, respectively, according to the post hoc power analyses. A larger sample size will be needed to clarify the correlation between BS and US findings of the ATFL, suggesting that there may not be a tremendous influence of BS on US findings of the ATFL of young women.

GJL is more prevalent in women than in men [1,31], and it was suggested that GJL (BS ≥ 4) may have a different effect on the native lateral ankle laxity by sex [13]. However, the limited sample size of high-BS subjects could not draw a conclusion regarding this issue. Thus, the present study aimed to compare the stress US findings of the ATFL between high- and low-BS groups of women, providing a more detailed insight into the association between GJL and lateral ankle laxity. The results of the current study suggest that BS may not be an appropriate tool for evaluating the native lateral ankle laxity in young women. The BS was originally developed as a screening tool for GJL in large populations [4,32]. This system evaluates mainly joints of the upper limbs (6/9), not ankle joints. In addition, it has been confirmed that the BS is affected by sex, age and race [1,5,31]. Therefore, the appropriate cut-off value of the BS has not been defined, and a higher score is recommended to discern patients with and without GJL among hypermobile individuals, such as young girls [5,33,34]. As mentioned above, the presence of GJL affects the surgical strategy in the treatment of CLAI [8,9,29]. Thus, the accurate assessment of the GJL is an important matter. There is a lack of studies evaluating the influence of GJL on native ankle laxity [12], and there is a lack of evidence regarding the association between GJL and ankle joint laxity. Therefore, the best tool for detecting GJL and determining its correlation with lateral ankle laxity needs to be established. Further basic and clinical studies are needed to evaluate the relationship between GJL and native ankle laxity using various measures other than US, such as an arthrometer.

No specific tool for evaluating the native ankle joint laxity has yet been developed. The Lower Limb Assessment Score (LLAS), introduced by Ferrari et al. [35], is a measure to evaluate hypermobility in the lower limb. This assessment tool evaluates the range of motion of the hip, knee, ankle and rear, mid and forefoot. It was reported that a cut-off score of ≥7/12 could identify individuals with lower limb specific hypermobility, with a specificity of 86% and a sensitivity of 68% [35,36]. Given that the main parameters of this scoring system are composed of joints of the lower extremities, the LLAS may be more appropriate for identifying the presence of ankle joint laxity than the BS. Future studies may be interesting to investigate the usefulness of the LLAS for detecting ankle laxity. Song et al. reported that the ATFL height (degree of ligament loosening) on US had the strongest relationship with the BS (r = 0.76) and proposed the ATFL height as a possible alternative for BS for the assessment of ankle joint laxity in GJL [12]. We did not evaluate the ATFL height, therefore, the significance of the ATFL height was not clear in the present study. Recently, several authors have demonstrated the usefulness and validity of US for the diagnosis of CLAI [15,16,34]. US evaluation is noninvasive and easily performed in the clinical setting. If some of the US findings of the ATFL in the healthy ankle are shown to be well associated with GJL, these parameters may replace the BS for estimating the native lateral ankle laxity. Therefore, further studies may be needed to find out US findings, such as the ATFL height, that would serve as screening tools for the lateral ankle laxity in patients with GJL.

There are several limitations to the present study. First, we did not evaluate the relationship between the BS and lateral ankle laxity in healthy men or other age groups of women. This study also assessed a single ethnic population. Therefore, the generalizability of the results in this study to a different age cohort or other ethnic groups is questionable [1,2]. Second, it remains unclear how exactly the BS reflects the presence of GJL. We did not perform genetic testing for the definitive diagnosis of GJL [37,38]. However, the BS is the most commonly used and validated measure to evaluate GJL [2]. Therefore, it is valuable to understand the association between BS and lateral ankle laxity. Third, the lateral ankle laxity was evaluated using the stress US method reported by Yokoe et al. [13]. However, the ATFL ratio may differ when using other procedures. Fourth, US evaluation is affected by the examiner’s experience and spatial resolution of the US apparatus. Fifth, we did not evaluate the relationship between the BS and lateral ankle laxity in healthy men. Sixth, the activity level of the participants was not considered. Finally, we did not evaluate the possible effect of stretching exercises on ankle ligament laxity [39]. Despite these limitations, the strength of the present study is that the influence of BS on lateral ankle laxity was prospectively evaluated by comparing high-BS (≥6) and low-BS (≤3) groups.

## 5. Conclusions

No significant differences in the stress US findings of the ATFL were found between high-BS (≥6) and low-BS (≤3) groups of young female subjects. In addition, no correlation was detected between the BS and stress US findings of the ATFL. The results of this study suggested that the BS cannot be used as an equivalent alternative to test for lateral ankle laxity in young female subjects. Clinicians should be cautious when estimating the native lateral ankle laxity of this population using the BS.

## Figures and Tables

**Figure 1 jcm-11-01759-f001:**
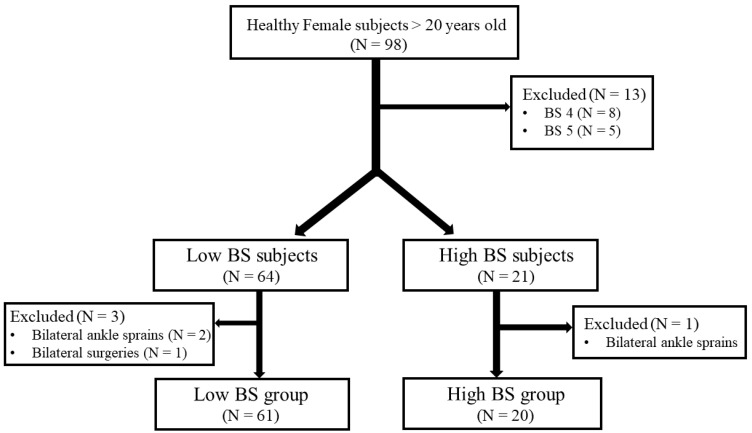
Flowchart of participant enrollment.

**Figure 2 jcm-11-01759-f002:**
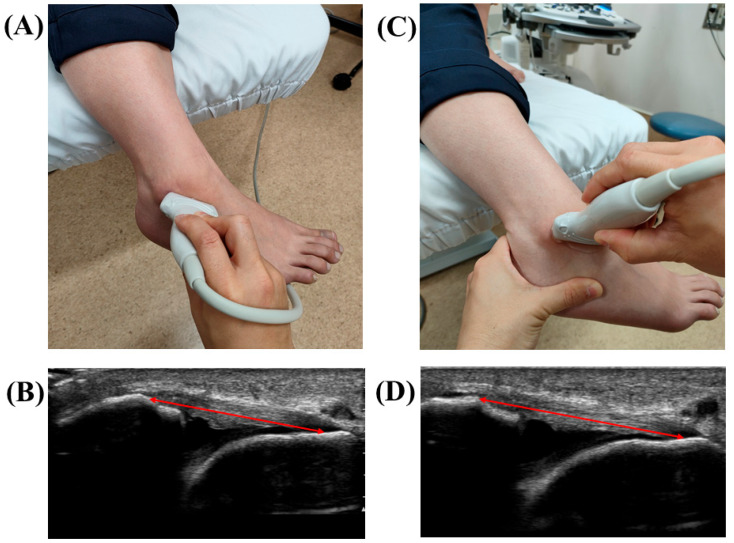
Ultrasonographic evaluation of the anterior talofibular ligament (ATFL). (**A**) Nonstress ATFL position. (**B**) An ultrasonographic picture of the nonstress ATFL. The red arrow shows ATFL length. (**C**) Stress ATFL position. Manual maximal internal rotation with varus talar tilt was applied to the ankle. (**D**) An ultrasonographic picture of the stress ATFL. The red arrow shows ATFL length.

**Figure 3 jcm-11-01759-f003:**
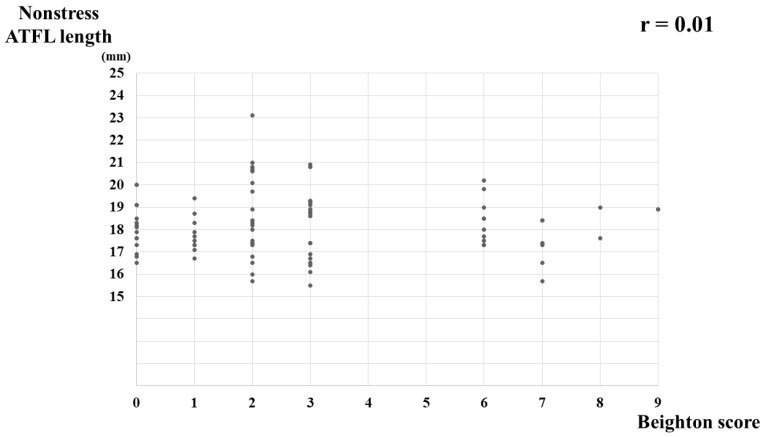
Correlation between the Beighton score and nonstress ATFL length in healthy young women. ATFL, anterior talofibular ligament.

**Figure 4 jcm-11-01759-f004:**
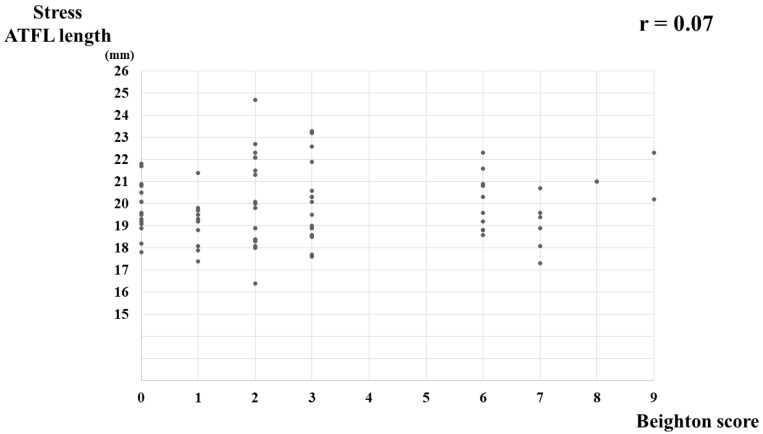
Correlation between the Beighton score and stress ATFL length in healthy young women. ATFL, anterior talofibular ligament.

**Figure 5 jcm-11-01759-f005:**
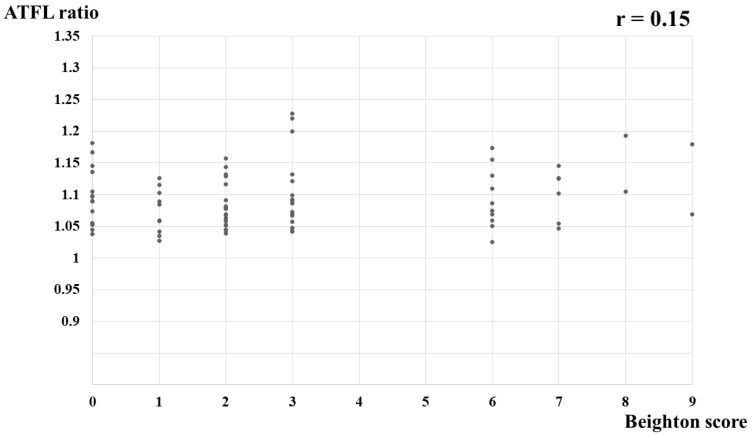
Correlation between the Beighton score and ATFL ratio in healthy young women. ATFL ratio, stress ATFL length/nonstress ATFL length; ATFL, anterior talofibular ligament.

**Table 1 jcm-11-01759-t001:** Participant chracteristics of the two groups.

Variables	Low-BS Group (*n* = 61)	High-BS Group (*n* = 20)	*p* Value
Age, year	23.7 ± 2.1 (20–33)	24.3 ± 1.8 (21–27)	0.14
Height, cm	157.5 ± 5.7 (147.4–166.1)	159.5 ± 5.2 (147.0–167.2)	0.21
Weight, kg	50.3 ± 4.6 (41.0–60.4)	52.4 ± 4.8 (44.4–63.8)	0.09
BMI	20.3 ± 1.4 (18.4–25.2)	20.8 ± 1.5 (18.0–23.4)	0.13
Foot size, cm	23.0 ± 1.1 (21.0–25.3)	23.3 ± 1.1 (20.5–25.2)	0.22
Side of the ankle, *n* (%)			0.25
right	45 (73.8)	12 (60.0)	
left	16 (26.2)	8 (40.0)	
Beighton score	1.6 ± 1.1 (0–3)	6.8 ± 1.0 (6–9)	<0.001

Data are shown as means ± standard deviations unless otherwise indicated. The number in the parenthesis shows range. BS, Beighton score; BMI, body mass index; cm, centimeter; kg, kilogram.

**Table 2 jcm-11-01759-t002:** Results of ADT and ultrasonographic evaluation.

Variables	Low BS Group (*n* = 61)	High BS Group (*n* = 20)	*p* Value	Effect Size
ADT			0.03	0.60
Grade 1, *n* (%)	44 (72.1)	9 (45.0)		
Grade 2, *n* (%)	17 (27.9)	11 (55.0)		
Ultrasonographic findings				
nonstress ATFL length, mm	18.2 ± 1.5 (17.8–18.6)	18.1 ± 1.1 (17.6–18.6)	0.99	0.08
stress ATFL length, mm	19.9 ± 1.7 (19.4–20.3)	20.0 ± 1.4 (19.3–20.6)	0.58	0.06
ATFL ratio	1.09 ± 0.05 (1.08–1.10)	1.10 ± 0.05 (1.08–1.13)	0.19	0.20

Data are shown as means ± standard deviations unless otherwise indicated. The number in the parenthesis shows a 95% confidence interval. ATFL ratio: stress ATFL length/nonstress ATFL length. BS, Beighton score; ADT, anterior drawer test; ATFL, anterior talofibular ligament.

## Data Availability

The data presented in the present study are available on request from the corresponding author.

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
