# Peer review of "Association between the Beighton Score and Stress Ultrasonographic Findings of the Anterior Talofibular Ligament in Healthy Young Women: A Cross-Sectional Study"

_jcm, 2022, doi:10.3390/jcm11071759_

Round 1
Reviewer 1 Report
Dear Authors,
Recommendations are shown in the file.
Best Regards

Reviewer 2 Report
General Comments
Thank you for the opportunity to review this study assessing generalized joint laxity in uninjured young women and its relationship to anterior talofibular ligament laxity. The study represents a good step in addressing the presence of laxity at a specific ligament that is frequently injured, and the potential influence or interaction that generalized laxity may have as a step toward improving post-surgical outcomes. The authors provide a clear and well-organized paper. The manuscript could benefit from some additional rationale/justification, clarification and details to help reader comprehension. I have some concerns about the statistical analyses selected, as well as the power and effect size observed with this sample.
I’m not sure if the journal requires it, but please consider applying the STROBE checklist to ensure all parts are included in the manuscript as recommended. I believe the authors have met most of the points, but some may be helpful to include, specifically item 5, 7, 9, and 12 in greater detail.
Specific Comments
Introduction
Page 2
Line 48 Please check reference order
Lines 44-54 The introduction is relatively short and readers would benefit from sentences addressing the following: What is the rationale for including only females? What is the rationale for using a higher BS cutoff (6 instead of 4 or 5)? What will the findings provide clinicians/surgeons/patients? The authors do an excellent job of providing rationale/justification on page 8 lines 222-228. Some version of those sentences would strengthen the introduction.
Methods
Line 63 Please clarify if Unilateral or Bilateral ankle injury was an exclusion criteria? Bilateral is mentioned in figure 1 and later in the text. Would a unilateral injury indicate the other ankle was included? This is unclear.
Line 72 I appreciate that no clear BS cutoff exists, and most studies to date have used either 4 or 5. My concern is that the change to ≥6 makes the results of this study less comparable to existing literature. If previous studies found significant differences, why make this change? Have the authors considered re-running their stats and seeing if the results change? That may provide important evidence in interpretation and clinical implications and would be a relevant finding in and of itself.
Page 3
Line 86 Would “gender” be a more consistent and appropriate term here than “sex” as gender is used previously?
Line 90 and Line 94 I appreciate the authors citing previous reliability studies, and that the surgeon was senior and blinded to BS. However, it would strengthen the paper and analysis to indicate if reliability testing was performed to establish intra-rater reliability for the ADT and US measures. While literature values are helpful, I believe there is a skill and consistency level associated with these measures that is helpful for readers to see established by the specific research team performing them.
Line 93 Could the authors define how the “dominant” side was defined? And if the participant had a unilateral ankle injury history, the non-injured side was tested? Please clarify.
Page 4
Line 124-126 Did the authors standardize or otherwise measure the force applied and the speed with which the force was applied? I understand participants were told to relax, and EMG was not used, which would have ensured no muscle activation. But how could the research team know a consistent load was applied to the ankle during US testing? This may have greatly impacted results. Establishing reliability would prove useful here as well.
Line 111-135 The authors cite previous study/studies for US measures, but greater detail is needed in this section. Pictures were taken, but how was the timing of pictures determined? And was it a static shot or a video recording that was screen captured? What software was used to measure the distance during resting and stress? Establishing rater reliability for landmark identification and length measurements would be useful here.
Lines 139-144 Please clarify which tests were used on which variables. My impression is that ADT is categorical, and would not be normally distributed, while ATFL length and ratio may be as a continuous variable. And BS also seems potentially not continuous and therefore not normally distributed as categories were used and certain scores excluded?
Page 5
Results
Line 154 Please indicate the results of the Shapiro-Wilk test here. What results influenced decisions on which statistical tests to run on which variables?
Line 164 These are exceptionally low r-values, so the r-squared will be very low. Do the authors have concerns about power here? While the a-priori power analysis seems adequate, I am wondering if something was missed? Possibly with the differences in sample sizes (especially compared to cited references that have more equal group samples) or the change in BS criteria.
Page 7
Discussion
Line 183 I appreciate the role of CLAI in the discussion and rationale for the study. I would like to see some brief discussion regarding the potential interaction of injury and GJL. This study focused on uninjured controls, and I am wondering what the impact of chronicity is in this population that may not be represented in this sample of participants?
Line 197 I think a discussion of the power and effect size for the analyses completed in this paper is warranted. I believe the participant ratio from the a-priori analysis was 1:3, which seems different from cited research, which might be closer to 1:1. What effect did this have? What effect did the shift in BS criteria have on results? There is a substantial overlap in the ranges reported. Were there a few outliers in other studies potentially?
Line 245 I think one more sentence is needed here to help the reader understand why this should be done. What is the end result the authors are looking to achieve? Why do we need to screen for lateral ankle laxity in patients with GJL, particularly if they are healthy or uninjured?
Line 258 Thanks to the authors for mentioning not including men. Can this be addressed further, possibly earlier in the rationale? What was the purpose? GJL is more common in females? Females have greater ankle sprains/instability? Something else?
Page 9
Conclusions
Line 266 I’m not sure it goes here, but a discussion of the power and effect sizes from the results may be beneficial to help readers contextualize findings. Were there no statistically significant differences or was the study underpowered?
Table 2 Please include either power or effect size values for comparisons.
Reviewer 3 Report
Comments to the Manuscript ID jcm-1622404 titled “Association between the Beighton score and stress ultrasonographic findings of the anterior talofibular ligament in healthy young women: A Cross-Sectional Study”. This research analyze the change in the anterior talofibular ligament in hyperlaxity women and health using different test laxity. The manuscript is well structured and written properly. Minus grammar mistakes but major changes are required.
Line 15: What means the symbol ÷?
Keywords: Are Mesh terms?
- Introduction
Lines 36-37: Please rewrite this sentence better, erase the term “why”.
Lines 36-39: If the BS not reflect the ankle laxity…is this a reliable test in your research, so…your research is reliable?
Lines 48-50: Why authors have include only women in the research?
Lines 57-59: The hypothesis is declare once the results have been obtained?. The normal hypothesis is opposite.
- Material and Methods
- Did authors include or exclude different foot types as cavus, valgus, flatfoot…. Think authors the foot type can influence in the ATFL?
Lines 69-70: “no evidence-based…”: Is your research reliable?
Lines 73-76: This paragraph suits better in Statistical analysis. Can authors explain the variability in the sample size between the high and low group? 20 to 61.
Line 78: Please correct “years” by “years”
Line 79: What means CI? Please explain.
Table 1.- Please add abbreviations and explain
Lines 86-89: All the feet were normal? There was not cavus or flatfoot?
90-96: Can authors add photos about the performance?
Lines 94-95: This must be in results.
Lines 99-103: Please explain better the scoring system
Lines 111-135: Can authors add photos?
Line 139: Please explain what mean CI
Table 2. Please add abbreviations (mm)
- Results OK
- Discussion
Lines 210-212: Based in this affirmation, is the research reliable? The sample size is not homogeneous.
Line 217-218: There is not a BS in ankle…so, how authors have calculate the score values?
Line 229: Once again authors exposed that the research is empirical.
Line 247: Authors constantly allude about a lot of limitations.
The conclusions are empirical based in a not reliable research.
Round 2
Reviewer 2 Report
Thanks to the authors for responding to reviewer comments. For the most part, the authors have made every attempt to address the limitations and add helpful information to the manuscript. I do have some continued concerns, particularly regarding reliability and power/effect size analyses. Clarifying these aspects for readers would help with transparency and interpretation of the strength of the findings.
Line 160 I appreciate that the authors indicated intra-rater reliability was established. However, my concern is that the ADT is a categorical variable (nominal) rather than a continuous variable. ICC is appropriate for continuous variables, but I think Kappa may be more appropriate for categorical variable. Please see Gwet KL. Intrarater Reliability. In Wiley Encyclopedia of Clinical Trials. 2008. Wiley & Sons Inc. Section 2. Nominal Scale Score Data available here https://www.researchgate.net/publication/227577647_Intrarater_Reliability
Please also see Haley SM, Osbert JS. Kappa Coefficient Calculation Using Multiple Ratings Per Subject: A Special Communication. Physical Therapy. 1989;69(11):970-974 for example. There are many other publications detailing Kappa usage as well. A statistical review may be needed.
Line 198 Thanks to the authors for including the rater reliability for US. I believe ICC is an appropriate measure for the ATFL variables. Ideally, ICC results are reported with the model type [eg. (2,1) or (3,1) or (2,k) or (3,k)] as well as the Standard Error of Measurement (SEM). See Shrout, P. E., & Fleiss, J. L. (1979). Intraclass correlations: Uses in assessing rater reliability. Psychological Bulletin, 86, 420-428 as well as
Denegar CR and Ball DW. Assessing Reliabilty and Precision of Measurement: An Introduction to Intraclass Correlation and Standard Error of Measurement. Journal of Sport Rehabilitation. 1993;2:35-42.
Lines 287 Referring to the original submission’s Line 164 comment: I appreciate the authors performed an apriori power analysis. However, I disagree that the obtained results would therefore have “guaranteed” power. The apriori analysis indicates what you can expect to find, based on pilot data or previous studies, with some degree of error. The power analysis indicated 80% power, which is appropriate, to detect a difference at a 2-tailed alpha level of 0.05. This is not foolproof, however, because if the study was re-run 99 more times, you would expect non-significance in at least 5 of those re-run studies. Additionally, was the correlation analysis the subject of the apriori power analysis, or was it based on ATFL length by Song? Power may change based on the variable measured. It would be helpful to readers to report the effect size and power associated with the correlation measures you report. I believe G*Power can calculate some of these using the Exact Test Family, but a statistical review may be needed.
Reviewer 3 Report
Authors have realized all the changes required by the reviewer. Congratulations.
Author Response
Thank you for your reviewing my revised paper. I am glad to hear that you have considered my revised paper acceptable.